# Clinical Efficacy of Brown Seaweeds *Ascophyllum nodosum* and *Fucus vesiculosus* in the Prevention or Delay Progression of the Metabolic Syndrome: A Review of Clinical Trials

**DOI:** 10.3390/molecules26030714

**Published:** 2021-01-30

**Authors:** Enver Keleszade, Michael Patterson, Steven Trangmar, Kieran J. Guinan, Adele Costabile

**Affiliations:** 1Department of Life Sciences, University of Roehampton, London SW15 4JD, UK; keleszae@roehampton.ac.uk (E.K.); michael.patterson@roehampton.ac.uk (M.P.); steven.trangmar@roehampton.ac.uk (S.T.); 2BioAtlantis Ltd., Tralee, V92 RWV5 Co. Kerry, Ireland; research@bioatlantis.com

**Keywords:** seaweeds, *Ascophyllum nodosum*, *Fucus vesiculosus*, metabolic syndrome

## Abstract

Metabolic syndrome (MetS) is a global public health problem affecting nearly 25.9% of the world population characterised by a cluster of disorders dominated by abdominal obesity, high blood pressure, high fasting plasma glucose, hypertriacylglycerolaemia and low HDL-cholesterol. In recent years, marine organisms, especially seaweeds, have been highlighted as potential natural sources of bioactive compounds and useful metabolites, with many biological and physiological activities to be used in functional foods or in human nutraceuticals for the management of MetS and related disorders. Of the three groups of seaweeds, brown seaweeds are known to contain more bioactive components than either red and green seaweeds. Among the different brown seaweed species, *Ascophyllum nodosum* and *Fucus vesiculosus* have the highest antioxidant values and highest total phenolic content. However, the evidence base relies mainly on cell line and small animal models, with few studies to date involving humans. This review intends to provide an overview of the potential of brown seaweed extracts *Ascophyllum nodosum* and *Fucus vesiculosus* for the management and prevention of MetS and related conditions, based on the available evidence obtained from clinical trials.

## 1. Introduction

Metabolic syndrome (MetS) is a collection of metabolic abnormalities that include conditions such as abdominal obesity, increased blood pressure (BP), increased fasting plasma glucose (FPG), increased triglycerides (TG) and decreased high-density lipoprotein cholesterol (HDL-C) that lead to an increased risk of developing cardiovascular diseases (CVDs), type 2 diabetes mellitus (T2DM) and all-cause mortality [1,2]. Metabolic syndrome has been one of the major public health challenges worldwide and it is estimated that approximately one-quarter of the world’s population is affected [3]. Excessive energy intake and lack of exercise result in a positive energy balance which leads to the accumulation of visceral fat, the progression of liver steatosis and the onset of MetS risk factors [4]. Since the prevalence of these metabolic dysfunctions is continuing to increase, the discovery of new strategies for the prevention or treatment of MetS risk factors is of importance [5,6].

The first-line of therapy for MetS is diet and lifestyle modifications including reducing caloric intake, adopting a healthy diet and increasing physical activity [7]. However, these approaches are often not sufficient and patients are commonly put on medications [8]. To date, the US Food and Drug Administration (FDA) has not approved any medication to treat MetS; however, an insulin-sensitizing agent, such as metformin, is currently widely administered in patients with MetS at the start of hyperglycemia treatment [9]. It has been also shown that metformin helps to reverse the pathophysiological alterations associated with MetS when it is administered in conjunction with lifestyle modifications [10] or with peroxisome proliferator-activated receptor agonists (PPARγ), such as thiazolidinediones and fibrates which promotes adipocyte differentiation and improve insulin resistance [11,12,13,14,15]. Although such medications can be helpful, most of them cause adverse effects and their effectiveness could be reduced or lost as a result of chronic administration [16]. Thus, there is emerging interest in the use of natural products to lower the risk and progression of MetS.

In recent years, marine organisms, especially seaweeds, have been highlighted as potential natural sources of bioactive compounds and useful metabolites, with many biological and physiological activities to be used in functional foods or in human nutraceuticals for the management of MetS comorbidities [17,18,19,20,21]. The major bioactive compounds of seaweeds are polysaccharides, in addition to phenolic, phlorotannins, terpenes, terpenoids, amino acids, proteins, peptides, lipids and halogenated compounds [22]. Among the various bioactive constituents, there is some evidence that some components in seaweed may have beneficial effects including anticoagulant [23], anti-inflammatory [24], antioxidant [25], anticarcinogenic [26] and antiviral activities [27]. However, the evidence base relies heavily on cell line and small animal models, with few studies to date involving humans.

Seaweeds are a widespread group of autotrophic organisms that have a long fossil history. They are globally distributed and can be located in every climatic zone ranging from freezing cold polar regions to tropical warm waters [28]. At present, more than ten thousand different species of seaweed are identified [29]. Seaweeds are classified into three main groups, namely red seaweeds (Rhodophyceae), brown seaweeds (Phaeophyceae) and green seaweeds (Chlorophyceae), each having specific nutritional, biological and chemical characteristics [30]. Among the three groups of seaweeds, brown seaweeds are known to contain more bioactive components than either red or green seaweeds [31]. The most abundant polysaccharides in brown seaweeds are laminarin, fucoidan and alginates [32].

Laminarins have been reported to exert bioactive properties in the gastrointestinal tract and are recognized as a regulator of intestinal metabolism through its impacts on mucus structure, intestinal pH and short chain fatty acids production [33]. Furthermore, laminarins provide protection against infection caused by bacterial pathogens and protection against severe irradiation, boosts the immune system by increasing the B cells and helper T cells and can also act on typical mechanisms involved in MetS, since they lower the systolic blood pressure, cholesterol absorption in the gut and consequently the levels of cholesterol and total lipid both in serum and liver [34,35].

Fucoidans have been reported to reduce hyperglycaemia via the inhibition of α-amylase and α-glucosidase, consequently decreasing intestinal absorption of glucose and enhancing the insulin-mediated glucose uptake due to the ability of fucoidans to modulate relevant pharmacological targets including glucose transporter GLUT-4 and AMP-activated protein kinase (AMPK) [36]. Fucoidans have been also reported to increase the expression of hormone-sensitive lipase, the key enzyme involved in lipolysis which suggest that fucoidans decrease lipid accumulation by triggering lipolysis [37,38,39,40]. Moreover, fucoidans are recognized for their cardiovascular and antihypertensive effects through the inhibition of the angiotensin converting enzyme (ACE) and the activation of eNOS-dependent pathways [41].

Alginates have been shown to inhibit the digestive enzymes pancreatic lipase and pepsin and diminish the intestinal absorption of triacylglycerols, cholesterol and glucose [33,42,43]. It has been also shown that, as with other dietary fibres, the consumption of alginates could delay gastric emptying, increase digestive fluid viscosity and reduce calorie intake through enhanced satiety [44,45]. The mechanisms of these molecules in the management and progression of MetS are summarised in Figure 1.

Among different brown seaweed species *Fucus vesiculosus* (*F. vesiculosus*) and *Ascophyllum nodosum* (*A. nodosum*) are the most studied species with the highest antioxidant values and highest total phenolic content (TPC) along with the greatest DPPH (2,2-diphenyl-1-picryl-hydrazyl-hydrate) radical scavenging activities [46,47,48].

Thus, this review intends to provide an overview of the potential of brown seaweed extracts *A. nodosum* and *F. vesiculosus* for the management and prevention of MetS and related conditions, based on the available evidence obtained from clinical trials.

## 2. Search Strategy

A comprehensive search of literature was carried out using electronic databases including Clinical Trials.gov, Medline, PubMed, Science direct, Google scholar to identify relevant studies in August 2020. Criteria for inclusion in this review were: (1) human adults (aged 18 and over), (2) dietary brown seaweed intervention (either *Ascophyllum nodosum* or *Fucus vesiculosus*; or in combination), (3) included anthropometric parameters, inflammatory markers, glucose, insulin, blood lipids and energy intake as an outcome and (4) written in English. Owing to the small number of eligible papers, trials both with and without dietary restriction were included in spite of the potential for weight change to influence results and there was no limit placed on follow-up or study duration. Papers were omitted if they were not original research or if the study involved cell culture or animal models. An overview of the clinical trials included in the review are summarised in Table 1.

## 3. Review of Clinical Trials Exploring the Impacts of *A. nodosum* and *F. vesiculosus* for the Management and Prevention of MetS and Related Conditions

### 3.1. Impacts on Appetite

Appetite is a mental feeling of hunger, satiation, satiety and a desire to eat specific type of food and is one of the factors affecting calorie intake [58,59]. Hall et al. (2012) investigated the effects of consuming *A. nodosum* enriched bread (4% *A. nodosum* per 400 g loaf) as part of a meal on energy intake in otherwise healthy 12 males. As compared to control bread (0% *A. nodosum*), consumption of this enriched bread at breakfast led to 16.4% significant reduction in energy intake at a test meal 4 h later and reported to significantly lower 24-h total energy intake by 506.1 kcal. No significant differences were seen in glycaemic or cholesterolaemic factors following the administration of the *A. nodosum* enriched bread compared to the control bread, which suggested that neither delayed gastric emptying nor nutrient encapsulation occurred. Moreover, no significant differences in hunger or fullness were reported by these authors [50].

In contrast, Mayer and co-authors observed no differences in energy intake between active and placebo groups after 1-week treatment with 400 mg *A. nodosum*, 1000 mg Garcinia Cambogia and 40 mgL carnitine a day. Compared to placebo, active treatment resulted in significantly increased satiety and fullness ratings and reduced subjective hunger sensations. Concomitantly, authors reported that the active treatment was also associated with a reduction in implicit wanting and explicit liking for savoury foods and a reduction in the preference for high fat foods in both study groups [52].

### 3.2. Impacts on Controlling Blood Glucose Levels

Diabetes is metabolic disorder that is characterized by chronic hyperglycaemia resulting from disturbances in insulin secretion and tissue resistance to its action [60]. Dietary carbohydrates are the major source for blood glucose [61]. These carbohydrates are hydrolysed by pancreatic α-amylase, followed by α-glucosidase before being absorbed in the small intestine [62]. One practical approach for decreasing postprandial hyperglycaemia is to retard absorption of glucose by inhibiting carbohydrate hydrolysing enzymes, α-amylase and α-glucosidase, in the digestive organs [63]. Inhibition of the two intestinal enzymes has been documented to significantly attenuate the increase of blood glucose levels after a mixed carbohydrate meal by delaying glucose absorption [36,46,64].

The first clinical trial exploring the antidiabetic properties of *A. nodosum* and *F. vesiculosus* was carried out by Paradis and colleagues in 2011. They studied the impact of brown seaweed blend containing *A. nodosum* and *F. vesiculosus* on plasma glucose and insulin concentrations over a period of 3 h post carbohydrate ingestion at pre-specified time points in 23 healthy subjects. Compared with placebo, consumption of 500 mg brown seaweed led to a significant 12.1% reduction in the insulin incremental area under the curve and a 7.9% increase in the Cederholm index of insulin sensitivity. Acute intake of the brown the seaweed extract prior to a carbohydrate load had no significant effect on plasma glucose levels [49].

Murray and co-authors examined the impact of a single ingestion of two doses of *F. vesiculosus* extract (500 mg and 2000 mg) in 38 healthy adults 30 min before a 50 g of available carbohydrate from white bread. Compared with the placebo, neither dose had a lowering effect on postprandial glucose or insulin responses. This study indicated that healthy Asian adults have higher postprandial insulin response, without any sign of glucose tolerance, compared with non-Asian adults, which could not be enhanced by a single dose of administration [53].

The same research group further investigated whether *F. vesiculosus* extract (2000 mg) moderated postprandial glycaemia in the evening in 18 healthy adults. The results of this double-blind, placebo-controlled, randomized crossover trial showed no effects on postprandial glucose and insulin levels after a single administration of the algae extract when compared with placebos, in the group as a whole. However, when just female participants were analysed, peak blood glucose concentration was reported to be reduced following the administration of *F. vesiculosus* extract [56].

In response to the prolonged administration (6 months) of *A. nodosum* and *F. vesiculosus* with the addition of chromium picolinate significantly reduced plasma levels of glucose, insulin and homeostatic model assessment (HOMA) index, suggesting an improvement of insulin sensitivity status in 50 overweight and obese subjects [36]. In line with the results reported by De Martin et al. (2018); in 65 dysglycemic patients, Derosa and collaborators (2019) observed a reduction in glycated haemoglobin (HbA1c), FPG, PPG and homeostatic model assessment for insulin resistance (HOMA-IR) after 6 months of treatment with the exact same nutraceutical combination [55]. The authors speculated that reducing postprandial plasma glucose with the nutraceutical, possibly, leads to a less work for the β-cells, which, in turn, preserve β-cell function longer. Furthermore, in 175 Caucasian patients with type 2 diabetes, the same research group similarly observed significant reductions in HbA1c, FPG and PPG after 6 months of treatment with the same nutraceutical administered in De Martin et al. (2018) and Derosa et al. (2019a) [36,55,57]. According to authors, phlorotannins which are a major polyphenol found only in marine brown algae, are the main compounds associated with this effect. However, it is important to consider that chromium picolinate is an important dietary supplement used to manage diabetes and solo use of algae extract are yet to be fully explored [65]. As such, any effect observed when combining chromium picolinate with *A. nodosum*, may be due to the effects of the former rather than the latter.

Along with the inhibition of the two intestinal enzymes, α-amylase and α-glucosidase, extracts from the brown seaweeds have been also shown to inhibit dipeptidyl-peptidase-4 (DPP-4) and have the ability to stimulate incretin hormone secretion [66]. The incretin hormones, glucagon-like peptide (GLP-1) and glucose-dependent insulinotropic polypeptide (GIP) also serve as a potential therapeutic target [67,68,69]. Incretin hormones are insulinotropic intestinal hormones that stimulate secretion of insulin in a glucose-dependent manner [66]. These intestinal hormones are rapidly broken down by the enzyme, DPP-4 [70]. DPP-4 inhibition increases insulin secretion and reduces glucagon secretion, thereby lowering glucose levels in the blood [71,72,73,74]. Furthermore, incretin hormones were found to inhibit apoptosis and promote pancreatic β-cell proliferation, which can intensify production of insulin and increase β-cell mass [75]. Thus, inhibiting the activity of DPP-4 and increasing GLP-1 and GIP secretion is an important strategy for controlling hyperglycaemia in type 2 diabetes patients [66,76].

### 3.3. Impacts on Anthropometric Indexes

With regards to anthropometric parameters, Iacoviello and colleagues observed a significant reduction in both bodyweight and body mass index (BMI), with no difference between the treatment and placebo groups. There were no changes in waist and hip circumferences and their ratio, systolic and diastolic blood pressure and heart rate between active and placebo groups [51]. After 6-month administration of food supplement containing *Ascophyllum nodosum*, *Fucus vesiculosus* and chromium picolinate, De Martin and co-authors reported a significant decrease in waist circumference, indicating that most of the subjects (88% of men and 77% of women) had lost weight [36]. Similarly, in type 2 diabetic patients, Derosa and colleges (2019) found a significant reduction in waist circumference in the intervention group compared to the placebo group after 6 months of intervention but no variation of body weight and BMI were recorded with the exact same nutraceutical combination administered in De Martin et al. (2018) [57]. Unlike previous studies that have reported improvements in various anthropometric parameters, in dysglycemic patients Derosa et al. (2019) did not report any significant changes in weight, BMI, waist, hip and abdominal circumferences after 6 months of treatment with the same nutraceutical combination that had been administered in the study by De Martin et al. (2018) and Derosa et al. (2019) [36,55,57]. Again, the effects reported in the above studies may be due to chromium picolinate rather than the macroalgae assessed.

### 3.4. Impacts on Blood Lipids

Iacoviello and colleagues (2013) observed a 5% significant reduction in triglyceride (TG) levels in the active treatment group compared to an increase of 2% in the placebo group at the end of 6-week intervention period in healthy adult subjects. The effect was not evident after 3 weeks of treatment, suggesting that at least 6 weeks of supplementation is required for it to be evident. In both treatment groups, a non-significant trend to reduced total cholesterol (TC) and low-density lipoprotein-cholesterol (LDL-C) and to increased high-density lipoprotein-cholesterol (HDL-C) was reported [51]. More recently, in contrast to afore mentioned findings, results from Derosa et al. (2019) showed no significant differences in TC, LDL-C, HDL-C and TG levels between baseline and 6 months post-treatment. However, it is important to note that the participants involved were type 2 diabetic patients with the majority of them under hypoglycaemic drugs and in particular receiving metformin [57].

### 3.5. Impacts on Inflammation

The link between the MetS and inflammation is well documented. The increased production of proinflammatory cytokines including C-reactive protein (CRP), tumour necrosis factor α (TNF-α) and interleukin-6 may reflect an overproduction by an expanded adipose tissue mass [77,78,79,80]. Phlorotannin-rich extracts from brown seaweeds may provide a potential means of controlling inflammation by different mechanisms including inhibition of release of proinflammatory cytokines including TNF-α and IL-1β and IL-6 in vitro [54].

In 43 healthy subjects, Iacoviello et al. (2013) did not find significant differences between the treatment and placebo groups regarding soluble markers of inflammation (TNF-α, IL-6 and CRP) after 6 weeks of treatment with 1800 mg *A. nodosum* and 350 μg iodine [51]. In another randomized, double blind, placebo-controlled crossover trial Baldrick and co-authors (2018) reported that consumption of 100 mg *A. nodosum* polyphenols for a period of 8 weeks resulted in a modest (23%) decrease in lymphocyte DNA damage, but only in a subset of the total population who were obese. No significant changes in CRP, antioxidant status, or inflammatory cytokines were observed between the treatment and placebo groups [54]. Derosa et al. (2019) further evaluated the effects of nutraceutical combination containing polyphenols extracted from *A. nodosum, F. vesiculosus* and chromium picolinate on inflammation in dysglycemic patients. Unlike previous findings, Derosa and co-authors found an improvement in TNF-α and Hs-CRP levels. The changes in the level of cytokines were small but significant; this could be due to fact that the enrolled participants were not diabetic, but dysglycemic, where it has been previously demonstrated that cytokine concentrations are greater in diabetic compared to nondiabetic individuals [55,57].

### 3.6. The Effects of Brown Seaweed Extracts on the Gut Microbiota

Accumulating evidence reveals that the gut microbiota plays a crucial role in maintaining intestinal homeostasis and improving metabolic health [81]. A relationship has been documented between the gut dysbiosis (an imbalance of gut microbiota composition) and the development of obesity, insulin resistance and other characteristics of MetS [82,83,84,85,86]. The mechanisms by which the gut microbiome impacts host physiology are mediated through short-chain fatty acids (SCFAs; e.g., acetate, butyrate and propionate) which are the most abundant product of bacterial fermentation of undigested dietary fibres [87].

Recently, a loss of *Akkermansia muciniphila* has been reported to be related to obesity and metabolic syndrome [83,88,89]. Microbiota composition in high-fat diet mice supplemented with dietary fucoidan for sixteen weeks from *A. nodosum* showed an increase in the relative abundance of SCFAs producing bacteria including *Akkermansia*, *Alloprevotella*, *Bacteroides* and *Clostridiales vadin BB60* and attenuated metabolic syndrome that is induced by high fat diet through reduced body weight, fasting blood glucose, hepatic steatosis, systematic inflammation and reduced insulin resistance [90].

To date, only one clinical trial has examined the administration of *Akkermansia muciniphila* and this three-month proof-of-concept study showed improvements in several metabolic parameters including body weight, fat mass, hip circumference, blood markers for liver dysfunction and inflammation [91]. Overall, brown seaweed extracts such as *A. nodosum* extract may have potential prebiotic activity by changing the composition and increasing the abundance of gut microbiota that helps to alleviate features of MetS [92,93]. In this instance, future clinical intervention trials with an appropriate design are warranted to explore the effects of brown seaweed extracts on gut microbiome composition.

## 4. Impact of Seasonal Variation and Extraction Techniques on Phenolic Content

The seasonal variation in the phenolic content of the brown seaweed extracts has been reported previously and, due to the potential use of *A. nodosum* and *F. vesiculosus* extracts in functional foods or in human nutraceuticals, determination of the most favourable time for harvesting the algal material is of importance and should be monitored to help standardise the finished products [17,62,94,95]. The metabolic production of polyphenolics relies on the harvesting season and location [95]. Indeed, *A. nodosum* collected from Norway had the highest polyphenolic content in winter season, while those collected from the Scottish west coast exhibited the highest phenolic content in July [94,95]. A similar pattern was observed by Apostolidis et al. (2011) in the *A. nodosum* collected from the Northeast U.S. Atlantic coast with the highest phenolic contents observed one in summer (June and July) and one in fall (October) [62].

There are various other factors that influence the production of phenolic metabolites in seaweeds such as severe defoliation, nutrient stress and environmental stress [62]. For the cold-water loving *A. nodosum* water temperature could be also a stress factor. This could potentially reveal the phenolic peak seen in the summer months, since it might be possible that under stress more phenolic metabolites are produced. The phenolic peak seen in October could be because of other environmental stress factors such as wave exposure, salinity, temperature and light intensity [96,97,98,99,100,101].

There were also species related variations in the carbolytic enzyme inhibitory activities by fucoidan isolated from *A. nodosum* and *F. vesiculosus* [63]. Depending on the target enzyme and collection period, fucoidan inhibited α-amylase and α-glucosidase activities differently. Fucoidan obtained from *A. nodosum* inhibited both α-amylase and α-glucosidase, whereas, fucoidan from *F. vesiculosus* is only effective against α-glucosidase [102,103]. Fucoidan from *A. nodosum* was shown to reduce the α-amylase activity between 7% and 100% at 5 mg/mL with IC50 values of 0.12 to 4.64 mg/mL based on the harvesting period. This inhibitory difference was mainly attributable to the chemical structure and the molecular weight of the fucoidans isolated from these two species [103]

Moreover, other experimental procedures and extraction methods might also affect the types of compounds isolated which may describe the differences in various compounds isolated from the same species of seaweed [103,104]. It is likely that different extraction and processing methods will have significant impact on the biological effects of these extracts in vivo, which may account for the disparities and inconsistent effects observed when comparing the results of clinical trials to date.

## 5. Limitations and Reported Adverse Events

The limitations of this review include a small number of eligible clinical trials, indiscriminate eligibility criteria and heterogeneity of methodologies.

None of the clinical trials included in the review reported any major adverse effects in response to the administration of seaweed extracts. Administration of *A. nodosum* and *F. vesiculosus* was well tolerated and there were no signs of organ toxicity or negative effects on physiological function. Notably, Iacoviello et al. (2013) did not observe any adverse consequences on thyroid function, an important finding when considering the iodine content of the brown seaweed extracts which has been previously documented to cause hyperthyroidism [51,105].

The currently available α-glucosidase inhibitors including acarbose, miglitol and voglibose produce gastrointestinal side effects, such as flatulence and diarrhoea, due to the fermentation of undigested carbohydrates in the intestine [106]. Paradis et al. (2011) showed that a relatively small dose of a α-amylase and α-glucosidase inhibitors from a brown seaweed extract was not accompanied by gastrointestinal intolerance or discomfort [49].

## 6. Conclusions

This is the first review to provide a comprehensive overview of the two most studied brown seaweed extracts *A. nodosum* and *F. vesiculosus* in the management and prevention of MetS and related conditions based on the available evidence obtained from clinical trials. Accumulating evidence from clinical trials indicates that brown seaweed extracts may have a potential role as food supplements for MetS management. However, many of the effects observed to date are inconsistent and in order to be effective in MetS management, seaweed extracts must become more clearly defined in terms of composition, extraction methods and a range of biological effects in vivo. Moreover, further clinical trials will be warranted to confirm any positive effects within different population groups and to establish the optimal dosage, duration of treatment, efficacy and safety.

## Figures and Tables

**Figure 1 molecules-26-00714-f001:**
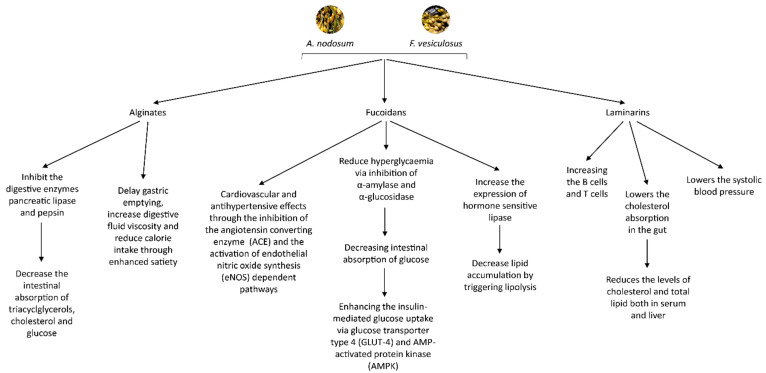
Summary of the molecules extracted from A. nodosum and F. vesiculosus in the management and progression of MetS.

**Table 1 molecules-26-00714-t001:** Summary of clinical trials included in this review (*n* = 10).

Authors	Subjects	Substance	Dosage	Study Design	Study Duration	Dependent Variables	Results
Paradis et al., (2011) [49]	23 healthy subjects (11 men,12 women), aged 18–60 years (mean age: 39 ± 12.7 years; mean BMI: 24.9 ± 3.2 kg/m^2^)	*Ascophyllum nodosum* and *Fucus vesiculosus* (a)	Two 250 mg seaweed capsules 30 min prior to the consumption of 50 g of carbohydrates from bread	Double-blind, randomized, placebo-controlled crossover study	3 h after the ingestion of the capsules	Plasma glucose and insulin concentrations	Compared with placebo, consumption of seaweed was significantly associated with a 12.1% reduction in the insulin incremental area under the curve and a 7.9% increase in the Cederholm index of insulin sensitivity. The single ingestion of brown seaweed had no significant effect on the glucose response
Hall et al., (2012) [50]	12 overweight or obese men, aged 18 and 65 years (mean age: 40.1 ± 12.5 years; mean BMI: 30.8 ± 4.4 kg/m^2^)	*Ascophyllum nodosum* (b)	100 g of bread containing *Ascophyllum nodosum* (4%)	Single blind crossover study	4 h after administration of *Ascophyllum nodosum*	Energy intake, appetite, plasma glucose and insulin	Significantly reduced energy intake at a test meal 4 h following administration of *Ascophyllum nodosum* bread. No changes in plasma glucose and cholesterol levels
Iacoviello et al., (2013) [51]	43 healthy subjects (19 men, 24 women), aged 21–63 years (mean age: 45.75 ± 9.51 years; mean BMI: 28.2 ± 4.8 kg/m^2^)	*Ascophyllum nodosum* and iodine (a)	Two capsules of algae, each containing 900 mg algae and 175 μg iodine	Randomised, placebo-controlled, double-blind trial, following a crossover design	14 weeks (6-week A. nodosum or placebo, 2 weeks of washout, additional 6-week with other treatment)	Anthropometric indexes and biomarkers of metabolic risk for cardiovascular disease	Significant decrease in both body weight and BMI, with no difference between the treatment and placebo groups. Compared to placebo group, Triglyceride levels were significantly lowered by 5% in treatment group after 6 weeks of active treatment. A non-significant trend to decreased total cholesterol and low-density lipoprotein and to increased high-density lipoprotein was observed in both supplementation groups. TNF-⍺ was significantly increased in the placebo group, but not in the active group, while adiponectin was significantly increased in both groups, with no difference between them
Mayer et al., (2014) [52]	28 healthy subjects (7 men, 21 women), aged 18–45 years (mean age: 31 ± 5 years; mean BMI: 22.6 ± 1.7 kg/m^2^)	*Ascophyllum nodosum, Garcinia cambogia* and L-carnitine (a)	Two capsules a day, each containing 200 mg *Ascophyllum nodosum*, 500 mg *Garcinia cambogia* and 20 mg l-carnitine	Double-blind, prospective, randomized, cross-over, placebo-controlled pilot study	1 week	Satiety sensations and food preferences	No differences in energy intake between study groups. Active treatment significantly reduced subjective hunger sensations and significantly increased satiety and fullness ratings
Murray et al. (2018) [53]	38 healthy subjects (9 men, 29 women), aged 19–56 years (median 23 years), BMI18.9 to 28.3 kg/m^2^ (median 21.9 kg/m^2^)	Fucus vesiculosus (a)	Each participant consumed a low dose (500 mg) *Fucus vesiculosus*, a high dose (2000 mg) *Fucus vesiculosus* and placebo (2000 mg cellulose) 30 min prior to 50 g of available carbohydrate form white bread	Double-blind, placebo-controlled, randomised cross-over study	2 h following carbohydrate consumption	Postprandial blood glucose and plasma insulin concentrations	No lowering effect on postprandial glucose or insulin responses. Different insulin sensitivity in Asian subjects
Baldrick et al. (2018) [54]	80 overweight/obese adults (39 men, 41 women), aged 30–65 years (mean age: 42.7 ± 7.1 years; mean BMI: 30.2 ± 3.9 kg/m^2^)	*Ascophyllum nodosum* (a)	A 400 mg capsule containing 100 mg *Ascophyllum nodosum* and 300 mg maltodextrin	Randomized, double-blind, placebo-controlled crossover study	8-week	DNA damage, plasma oxidant capacity, C-reactive protein (CRP) and inflammatory cytokines	Modest decrease in DNA damage but only in a subset of the total population who were obese. There were no significant changes in CRP, antioxidant status, or inflammatory cytokines
De Martin et al., (2018) [36]	50 patients (18 men, 32 women), aged 18–60 years (mean age: 54 ± 12 years)	*Ascophyllum nodosum, Fucus vesiculosus* and chromium picolinate (a)	Three capsules a day, each containing 237.5 *Ascophyllum nodosum,* 12.5 mg *Fucus vesiculosus* and 7.5 μg chromium picolinate	Observational study	6-months	Waist circumference, fasting blood glucose, HOMA index and insulin levels	Waist circumference decreased significantly after 6 months of treatment. Both blood glucose and insulin levels were significantly reduced after 6 months of treatment. HOMA index decreased significantly, suggesting an improvement of insulin sensitivity status
Derosa et al., (2019) [55]	65 dysglycemic patients (33 men, 32 women), aged ≥ 18 years (mean BMI: 28.9 ± 2.7 kg/m^2^)	*Ascophyllum nodosum, Fucus vesiculosus* and chromium picolinate (a)	Three capsules a day, each containing 237.5 *Ascophyllum nodosum,* 12.5 mg *Fucus vesiculosus* and 7.5 μg chromium picolinate	Double-blind, randomized, placebo-controlled study	6-months	Body Weight, Body Mass Index, Waist Circumference, Hip Circumference, Fasting plasma glucose (FPG), postprandial plasma glucose (PPG), HbA_1c_, fasting plasma insulin, HOMA index, high sensitivity C-reactive protein, tumour necrosis factor-α and adhesion molecules	FPG, PPG, HbA_1c_, HOMA-IR, CRP and TNF- α reduced significantly compared to placebo after 6 months. No significant changes were observed in anthropometric indexes
Murray et al., (2019) [56]	18 normotensive subjects (12 females, 6 males), aged 18–65 years (mean age: 25.5 ± 19 years; mean BMI 23.8 ± 2.6 kg/m^2^)	*Fucus vesiculosus* (a)	2000 mg powdered extract from *Fucus vesiculosus* containing 560 mg polyphenols and 1340 mg fucoidan	Double-blind, placebo-controlled, randomized crossover trial	3 h after ingestion of *Fucus vesiculosus*	Post prandial blood glucose levels	No effect on postprandial glycaemia. Only in females, peak blood glucose concentration was reduced after the polyphenol-rich extract
Derosa et al., (2019) [57]	175 Caucasian patients with type 2 diabetes (85 men, 90 women), aged 18 years and over (mean BMI: 27.5 ± 2.3 kg/m^2^)	*Ascophyllum nodosum*, *Fucus vesiculosus* and chromium picolinate (a)	Three capsules a day, each containing 237.5 *Ascophyllum nodosum,* 12.5 mg *Fucus vesiculosus* and 7.5 μg chromium picolinate	Multicentre, 6 months, double-blind, randomized, controlled, clinical trial	6-months	Anthropometric parameters (body weight, BMI, abdominal circumference), glyco-metabolic control (FPG, PPG, HbA_1c_) and lipid profile (Total cholesterol, TC; low-density lipoprotein-cholesterol, LDL-C; high-density lipoprotein-cholesterol, HDL-C and Triglycerides, TG)	No variation of body weight and BMI were recorded. A significant reduction of waist circumference was recorded in the nutraceutical group but not in placebo group. HbA_1c_, FPG and PPG were significantly reduced by the nutraceutical combination, but not by placebo. No variations of TC, LDL-C, HDL-C and TG were recorded compared to baseline.

The form of test substance administered in the clinical trials: (a) capsular form (b) powder incorporated into a meal.

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
