# Peer review of "Clinical Efficacy of Brown Seaweeds Ascophyllum nodosum and Fucus vesiculosus in the Prevention or Delay Progression of the Metabolic Syndrome: A Review of Clinical Trials"

_molecules, 2021, doi:10.3390/molecules26030714_

Round 1

Reviewer 1 Report

The authors have done a good job to comprehensively review the up-to-date clinical studies for brown seaweed on treatment of metabolic syndrome. It has provided a valuable reference for the researchers to learn more about advancement in this field. The authors have systematically described the effect of brown seaweed on  how it can improve the symptoms of metabolic syndrome including appetite, controlling blood glucose, anthropometric index, blood lipids, inflammation and also its side effects. It was well-recognized, however, I have some minor suggestions for the authors' consideration:

  1. Page 9: Section 3.6 is impressive as correlating to the most hot topic microbiota, but most of the content is not relevant to clinical study except the last paragraph line 264-271.  I suggest to remove the irrelevant content
  2. Section 4 (Page 9 line 272-Page 10 line 371) provided a good discussion on the seasonal variation of phenolic contents in brown seaweeds but I don't think it is directly related to the topic of this review

Author Response

Journal: Molecules (ISSN 1420-3049)

Manuscript ID: molecules-1087998

Type: Review

Number of Pages: 19

Title: Clinical Efficacy of Brown Seaweeds Ascophyllum Nodosum and Fucus Vesiculosus in the Prevention or Delay Progression of the Metabolic Syndrome: A Review of Clinical Trials

Authors: Enver Keleszade, Michael Patterson, Steven Trangmar, Kieran J. Guinan, Adele Costabile *

Comments of the Reviewers and Editor are in black, responses are in blue and citations from and changes in the manuscript are in red.

Reviewer 1 - Review Report (Round 1)

Comments and Suggestions for Authors

The authors have done a good job to comprehensively review the up-to-date clinical studies for brown seaweed on treatment of metabolic syndrome. It has provided a valuable reference for the researchers to learn more about advancement in this field. The authors have systematically described the effect of brown seaweed on how it can improve the symptoms of metabolic syndrome including appetite, controlling blood glucose, anthropometric index, blood lipids, inflammation and also its side effects. It was well-recognized; however, I have some minor suggestions for the authors' consideration:

  1. Page 9: Section 3.6 is impressive as correlating to the most hot topic microbiota, but most of the content is not relevant to clinical study except the last paragraph line 264-271. I suggest to remove the irrelevant content

Many thanks for your positive revision. We have addressed this point and the content that is not relevant to clinical study has been removed.

  1. Section 4 (Page 9 line 272-Page 10 line 371) provided a good discussion on the seasonal variation of phenolic contents in brown seaweeds but I don't think it is directly related to the topic of this review

Many thanks for raising this point. This comment has been also highlighted by reviewer 3. We felt to not remove this paragraph as the composition, extraction methods, optimal harvesting time and location should be standardized in further clinical trials to establish the optimal dosage, efficacy and safety of this products.

Reviewer 2 Report

The manuscript entitled "Clinical Efficacy of Brown Seaweeds Ascophyllum Nodosum and Fucus Vesiculosus in the Prevention or Delay Progression of the Metabolic Syndrome: A Review of Clinical Trials" is an overall well-structured manuscript as a review study.

As the authors' suggestion, the effects of seaweeds on metabolic syndrome in humans are not yet consistent.

And there is a typo in the author's name in the manuscript, so it needs to be corrected. In the manuscript, "Lacoviello" should be amended to Iacoviello.

Author Response

Reviewer 2 - Review Report (Round 1)

Comments and Suggestions for Authors

The manuscript entitled "Clinical Efficacy of Brown Seaweeds Ascophyllum Nodosum and Fucus Vesiculosus in the Prevention or Delay Progression of the Metabolic Syndrome: A Review of Clinical Trials" is an overall well-structured manuscript as a review study.

As the authors' suggestion, the effects of seaweeds on metabolic syndrome in humans are not yet consistent. And there is a typo in the author's name in the manuscript, so it needs to be corrected. In the manuscript, "Lacoviello" should be amended to Iacoviello.

We apologies for this mistake and we have amended accordingly.

Reviewer 3 Report

20th January 2021

Manuscript ID: molecules-1087998

Clinical Efficacy of Brown Seaweeds Ascophyllum Nodosum and Fucus Vesiculosus in the Prevention or Delay Progression of the Metabolic Syndrome: A Review of Clinical Trials

Comments to the Authors:

The purpose of the authors was to provide an overview of the potential use of brown seaweeds extracts Ascophyllum nodosum and Fucus vesiculous for the prevention and management of metabolic syndrome and comorbidities. They had explain that these brown seaweed are a source of bio compound such as antioxidants of polysaccharides such laminarins, fucoidans and alginates that are protector molecules involved in preventing the progression of the metabolic syndrome. Authors have also assessed an exhaustive search of investigations about the use of brown seaweed as nutritional supplementation to improve the metabolic state (weight loss, reduced glucose or insulin levels, reduced systolic blood pressure…). Moreover, they have discussed results and they also have highlighted the impact of these brown seaweeds on the improvement of the metabolism and on the inflammation state. However, authors stated that many of the effects observed to date are inconsistent and in order to be effective in MetS management, seaweed extracts must become more clearly defined in terms of composition, extraction methods and a range of biological effects in vivo. Also, they suggested that further clinical trials will be warranted to confirm the effects and to establish the optimal dosage, duration of treatment, efficacy and safety.

Minor points:

It is an interesting and well-crafted review.

However, some concerns have to be answered in order to improve the manuscript.

  1. It would be useful to introduce a paragraph with the current perspectives in metabolic syndrome treatment and on which pathogenic targets they act.
  2. Perhaps it would be interesting to add an explanatory figure of the mechanism of action of these molecules to facilitate understanding.
  3. It would be useful to include a paragraph of study’s main limitations.
  4. Please, review References. Someone is incorrect.

Author Response

  1. It would be useful to introduce a paragraph with the current perspectives in metabolic syndrome treatment and on which pathogenic targets they act.

We thank the reviewer for the positive revision. We have now added the current perspectives in metabolic syndrome treatment highlighting which pathogenic targets they act and now read as: To date, the US Food and Drug Administration (FDA) has not approved any medication to treat MetS; however, an insulin-sensitizing agent, such as metformin, is currently widely administered in patients with MetS at the start of hyperglycemia treatment. It has been also shown that metformin helps to reverse the pathophysiological alterations associated with MetS when it is administered in conjunction with lifestyle modifications or with peroxisome proliferator-activated receptor agonists (PPARγ), such as thiazolidinediones and fibrates which promotes adipocyte differentiation and improve insulin resistance. (Line number: 43 - 50)

  1. Perhaps it would be interesting to add an explanatory figure of the mechanism of action of these molecules to facilitate understanding.

We thank the reviewer for this comment. The Figure 1 now has been added.

  1. It would be useful to include a paragraph of study’s main limitations.

We thank the reviewer for this point and we have added the main study limitations and now read as: The limitations of this review include a small number of eligible clinical trials, indiscriminate eligibility criteria and heterogeneity of methodologies (Line number: 304 – 306).

  1. Please, review References. Someone is incorrect.

We thank the reviewer for this comment. This point has also highlighted by the reviewer 2 and we revised accordingly.